# Are Urban Populations of a Gliding Mammal Vulnerable to Decline?

**DOI:** 10.3390/ani13132098

**Published:** 2023-06-24

**Authors:** Anita J. Marks, Ross L. Goldingay

**Affiliations:** Faculty of Science and Engineering, Southern Cross University, East Lismore, NSW 2480, Australia; ross.goldingay@scu.edu.au

**Keywords:** small populations, mammal declines, population monitoring, squirrel glider, *Petaurus norfolcensis*, urban ecology, genetic rescue

## Abstract

**Simple Summary:**

Arboreal mammals provide important ecological services, but they are threatened by many anthropogenic activities, including habitat isolation and fragmentation. We investigated the population dynamics of the Australian squirrel glider in a small and larger reference population over a 16 year period. Population modelling suggests that a decline occurred in the small population but not the larger one. External factors are implicated in the decline and warrant further investigation.

**Abstract:**

Small populations are at high risk of extinction, and they are likely to need management intervention. Successful management, however, relies on sufficient long-term demographic data in order to determine whether apparent declines are natural fluctuations or the product of threatening processes. In this study, we monitored a small urban population of squirrel gliders (*Petaurus norfolcensis*) in Queensland, Australia, over a 16 year period. A reference population in a larger forest patch was also studied in order to investigate whether its demographic trends were similar. Using mark-recapture data to generate estimates of apparent survival and population size, we found evidence of a decline within the small population but not in the reference population over the monitoring period. We suggest that the influence of multiple factors may have led to the decline, but, ultimately, that the genetic condition of the small population may be responsible. Understanding demographic trends is an important context for management interventions of small populations, although causes of decline need to be identified for successful management. The squirrel glider provides a useful case study for small urban populations and particularly for arboreal mammals.

## 1. Introduction

Small populations are vulnerable to demographic and genetic impairment [1]. Demographically, they may be at risk due to unfavourable years, which cause reproduction to fail and adult mortality to increase. Genetically, small populations are at risk of inbreeding depression, an inevitable consequence of small, closed populations [2]. If environmental conditions are benign, small populations may be able to persist for long periods [3]. However, species of conservation concern should not be left to the stochastic forces of nature. 

Multiple factors lead to the occurrence of small populations: historical habitat clearing and fragmentation [4], the ravages of predators or disease [5,6], or over-exploitation [2]. Whatever the cause, species reduced to a small population size are at a high risk of extinction, and they are likely to need management intervention. Although the threats to small populations are well documented in conservation theory [7], there are benefits to collecting empirical evidence since variation can occur between the context and the taxon. Even for species that are not currently of conservation concern, investigating whether they persist at small population sizes may provide insights that can be applied elsewhere. 

Wildlife living in urban landscapes are particularly likely to have small populations due to urbanisation and fragmentation [8]. These populations may occur in a state of flux due to their ecological attributes and the fact that insufficient time may have elapsed for persistence to be resolved [9]. As a consequence, small urban populations can be viewed as model systems, which can reveal how species respond to reduced population sizes, usually in the context of a much more hostile surrounding matrix that limits dispersal. 

The Australian squirrel glider (*Petaurus norfolcensis*) provides one such example. This species has a geographic range that extends the length of eastern Australia and, consequently, overlaps areas that have undergone historic and recent urbanisation [10,11]. In south-east Queensland, the centre of its range, the squirrel glider is found across many urban forest remnants [12]. Maintaining this species in forest ecosystems may be highly beneficial because it is highly nectarivorous and is therefore involved in pollination [13,14], and its diet includes folivorous and sap-sucking insects that can cause tree damage [15,16,17,18]. In urban settings, the squirrel glider can tolerate some level of anthropogenic disturbance, but it still has specific habitat requirements [19]. However, its persistence over the long term is predicted to be contingent on active management [20,21]. The squirrel glider is closely related to two highly endangered species, the mahogany glider (*Petaurus gracilis*) and Leadbeater’s possum (*Gymnobelideus leadbeateri*), which are threatened by both habitat clearing and fragmentation [22,23,24]. Although these species have been well studied, the demographic consequences of having a small population size is currently unknown, and the squirrel glider provides an opportunity to obtain relevant insights for these species. 

We used mark-recapture data to model population and survival trends over 16 years in two squirrel glider populations in Brisbane, Queensland. Relying on just two populations is a limitation, of course, but the long period of detailed study provides insights that would not otherwise be available. Our purpose was to investigate the long-term viability of a small, urban population through comparison with a second peri-urban population that is not facing the same urban pressures. Long-term monitoring has been necessary to separate the inevitable year-to-year variation that occurs in wild populations from a sustained decline in those populations. This small, isolated urban population has been subject to increasing anthropogenic pressure and habitat loss, and its long-term persistence may be in question. The peri-urban population occupies a large but geographically fragmented forest patch, and it has not yet been subject to the same degree of anthropogenic pressure. We aimed to understand whether survival and population size have changed over time as a result of the demographic conditions of the two populations. We predicted that the urban population would decline over time, with limited growth potential due to its demographic structure and genetic diversity, its habitat size, and its lost connectivity. We predicted that the peri-urban population would be more stable over the long term given both the size of the forest patch and the lesser influence of urbanisation.

## 2. Materials and Methods

### 2.1. Study Area

This study was conducted at two remnant forest sites in Brisbane, Australia (Figure 1). The urban trapping grid (42 ha) was embedded in an urban matrix of housing estates and commercial businesses with some decreasing green corridors that attach it to a larger parkland in Cannon Hill (27°28′33.68″ S, 153°06′07.51″ E). The broader extent of forest suitable for the squirrel gliders at this location is approximately 100 ha, but it is less diverse and younger than on the trapping grid. This site has seen increasing fragmentation over the last fifty years, and it is now bounded by roads and houses. Further internal vegetation clearing and fragmentation has recently occurred due to the construction of new housing developments and a golf course. Dispersal into or out of the remnant is constrained by roads [11,20]. The peri-urban site was located in Burbank (27°31′40.43″ S, 153°08′02.15″ E), and this is a larger remnant forest patch (~700 ha) in a semi-rural landscape. The peri-urban trapping grid (25 ha) was a square grid embedded within the large forest remnant. The difference in the trapping grid size was due to the configuration of the habitat at the urban site, which is horseshoe-shaped with mostly hard edges. At each site over the course of the population monitoring, 30 trapping points were established within the trapping grids, with two traps installed at each point. 

Habitat conditions vary between the two study sites, and though larger in size, the peri-urban forest patch consists of less-productive flowering trees in comparison to the urban site. A large diversity of flowering eucalypts is present within the urban site, which has rendered it highly suitable for squirrel gliders. Additionally, squirrel gliders at the peri-urban site compete with a small number of sugar gliders (*P. breviceps*). 

### 2.2. Squirrel Glider Trapping

Squirrel gliders were captured across two multi-year periods in each population, with an intervening period of 6–9 years. At the urban site (hereafter, Site 1), trapping occurred mostly three times per year during 2004–2008 (referred to as years 1–5) and four times per year during 2015 (year 6) and 2017–2020 (years 7–10). At the peri-urban site (hereafter, Site 2), trapping occurred twice per year during 2004–2008 (years 1–5) and four times per year during 2018–2020 (years 6–8). Trapping occurred primarily between autumn and late spring in order to increase the likelihood of encountering females with pouch young. Trapping sessions within a year were spaced at least 1 month apart, whereas in consecutive years, they were separated by 6–9 months. 

Animals were captured in Elliott B traps (45 × 15 × 15 cm) installed on horizontal wooden planks attached to tree trunks that were 2–4 m above ground [25,26,27]. Each trap was secured to the wooden plank using two heavy rubber bands and encased in a plastic covering in order to protect it from the elements. Traps were baited with a mix of rolled oats, honey, and peanut butter. A diluted honey water mix was sprayed on the trunk above the trap as an attractant. 

Captured animals were weighed, sexed, and aged as either subadult or adult before release. Untagged animals were fitted with small numbered metal ear tags. Gliders were aged using a combination of tooth wear, vent colouration, and body weight, following methods previously described [25,26,28]. The reproductive condition of the captured females was assessed based on evidence of breeding, using pouch tautness, pouch young, and teat length to determine this, following methods previously described [28]. Activity of the male scent gland on the crown of the head was assigned a score according to maturity. 

### 2.3. Data Analysis

Capture histories were constructed from mark recapture data at the trapping grids within the two study sites. At Site 1, 34 sessions occurred between May 2004 and September 2020. At Site 2, 22 sessions occurred between June 2004 and September 2020. The Robust Design population model [29,30], as implemented in Program MARK (version 7.0) [31], was used to model the data. The model includes parameters for estimating apparent survival, capture and recapture probabilities, and temporary emigration. Survival is referred to as ‘apparent’ because true mortality cannot be separated from permanent emigration. An estimate of population size is derived from the parameters. To apply this model to our data, we treated a survey year as a primary sample period and the trapping sessions in each year as secondary samples. We therefore assume that the population is closed within a year but is potentially open across years. 

The two locations were analysed separately. Models were constructed to enable the two time periods for each location to be analysed together. Some simplifying assumptions were applied. The first was that survival could differ between time periods but was equal within a time period. No tagged animals survived the intervening period between time periods. We constructed models in which parameters could be year-varying or constant across a time period. If a model with year-varying capture parameters fitted the data better than one where they were constant, we re-ran the year-varying model with some years equal in order to reduce the complexity in the model. We excluded any models that did not converge. Preliminary analyses with models that included sex as a grouping factor revealed no better fit to the data than the models without sex, so this parameter was removed. Program MARK ranks candidate models from lowest to highest AICc value, with the Akaike Criterion Information corrected for a small sample size [32]. Models were compared based on the difference (Δ) in AICc values. Models were considered equally plausible if within 2 ΔAICc of the top ranked model, but less plausible as ΔAICc increases above 2 [32]. 

## 3. Results

### 3.1. Squirrel Glider Captures

At urban Site 1, 216 individual adult squirrel gliders were captured over 7217 trap-nights during 2004–2020, including 101 males and 115 females. At Site 2, 104 adults were captured over 4280 trapping nights during 2004–2020, including 41 males and 63 females. At Site 1, the sex ratio of captured gliders was biased towards females, being above parity (0.5) in all years except 2004 (0.43) (2004–2008 mean = 0.54, 2015–2020 mean = 0.61). The sex ratio was similarly biased towards females at Site 2, being above parity in all of the years (2004–2008 mean = 0.62, 2018–2020 mean = 0.61). 

### 3.2. Population Modelling and Demographics

The top model for the Site 1 population was highly supported with a model weight of 0.83, and it was 4.9 times more plausible than the next model (Table 1). In this model, there was no difference in annual survival across the two time periods, which was estimated as 0.32 ± 0.04. This model had the temporary emigration parameters set to zero. The probability of capture and recapture varied, but some years they were estimated with equivalent values. The estimate of initial capture was 0.27 ± 0.03 in years 1, 3, 4, 6, 8, 10; 0.42 ± 0.08 in year 2; and 0.54 ± 0.06 in years 5, 7, and 9. The estimate of recapture was 0.43 ± 0.03 in years 1–8, 0.22 ± 0.05 in year 9, and 0.81 ± 0.06 in year 10. 

The top model for Site 2 had a model weight of 0.92, which was 18.4 times more plausible than the second model (Table 1). Apparent survival for adult gliders was best explained as constant across the two time periods (0.70 ± 0.14), and the temporary emigration parameters were set to constant (g’ = 1.0, g” = 0.39 ± 0.07). Capture and recapture probabilities were equivalent, and they showed some variation throughout the years, with years 1–5 (0.68 ± 0.04) different from the later years (0.19 ± 0.07 in year 6, 0.48 ± 0.05 in year 7 and 0.59 ± 0.05 in year 8). 

### 3.3. Population Sizes

In 2004, the urban grid population at Site 1 was estimated at 70 individuals, and it increased to 94 in 2007, but declined in 2008 (Figure 2). In 2015, the population was smaller, with an estimate of 30 individuals. By 2020, the population was estimated at 27. The mean annual population size in each time period was 70 in 2004–2008 and 29 in 2015–2020. The estimates in 2017 and 2019 were close to the number of adults captured, whereas during 2004–2008 the estimates were 1.2–1.7 times higher than the number of individuals captured.

The population estimate on the peri-urban grid at Site 2 was consistently smaller than at Site 1 (Figure 2)., The population estimate was larger during 2018–2020 than in 2007–2008. The lower estimates in 2007–2008 may have been influenced by the lower trapping effort (3 nights per session) compared to other years (4–5 nights). The estimates were very close to the number of gliders captured during all years of sampling. 

## 4. Discussion

Urban populations can provide important insights into the dynamics of small populations [9]. In this study, we describe the dynamics of two populations of a gliding mammal––one that is small and isolated, and the other that is at least five times larger and interconnected. Generalising from just two populations requires caution, but population theory predicts that small populations will be vulnerable to collapse. Our population monitoring occurred over a period of 16 years, which provided an opportunity to properly identify the inevitable year-to-year fluctuations and conclude whether these populations showed evidence of stability, increase, or decline over this time. Our findings have relevance to several threatened species in similar scenarios around Australia, including the mountain pygmy possum (*Burramys parvus*) and the helmeted honeyeater (*Lichenostomus melanops cassidix*) [33,34].

### 4.1. Adult Population Size and Survival

We estimated the number of adult squirrel gliders on the two trapping grids. We estimated a mean of 70 adults in the first time period at Site 1 but only 26 adults at Site 2. The estimates are representative of the trapping grids, so it would be expected that the population sizes across the entire forested areas will be larger. The Site 2 forest patch is 28 times larger than its trapping grid, but the Site 1 patch is only two times larger than its grid. Therefore, the total population at Site 2 would be greater (28 × 25 adults) than at Site 1 (2 × 29 adults), despite the lower density at Site 2. Differences in the number of adult gliders on the grids can be explained by differences in habitat quality. Habitat type and quality are known to contribute to variation in local population densities [35,36,37]. Site 1 showed variation in population size in the first study period that was not observed at Site 2, which may reflect food resources more prone to annual variation in abundance. Site 1 contains 10 tree species that, in a good year, provide abundant nectar and pollen to gliders throughout the year [17]. However, pronounced year-to-year fluctuations in flowering have been observed at Site 1 [26]. This has been observed at other locations [38]. In contrast, Site 2 showed much less variation, which perhaps reflects the lower quality habitat (fewer mass flowering tree species) and lower variation in carrying capacity. 

In the second sample period, the mean size estimate was 29 adults at Site 1 and 25 adults at Site 2. This suggests that a decline had occurred at Site 1 but not at Site 2. Declines have been described in some populations of the Siberian flying squirrel (*Pteromys volans*) [39,40], but some uncertainty in the trend has remained due to the wide variation in annual estimates. In our case with the squirrel glider, we were able to characterise annual variation over 5-year blocks, which has provided us with confidence that a decline has occurred.

We have also been able to demonstrate clear differences across our two study areas in survival. Annual adult apparent survival was constant but low across the 5 year periods at Site 1 (0.32 ± 0.04, ±SE) but constant and much higher at Site 2 (0.70 ± 0.14). In one population of flying squirrels [40], annual apparent survival started at 0.62 ± 0.07 and finished at 0.34 ± 0.07 over an 11 year period. Annual adult survival in undisturbed populations of squirrel gliders has been estimated at 0.60–0.73 [41,42]. These findings suggest the low survival rate at Site 1 may be too low to sustain the population.

### 4.2. Causes of the Apparent Decline at the Urban Site

Identifying a decline leads to the question of the cause. The potential causes are fivefold: (i.) habitat decline or loss; (ii.) drought; (iii.) increased predation; (iv.) disease; and (v.) inbreeding. Some of these may have acted together. The decline in the population appeared to occur after the first survey period. However, there was no decline in habitat extent or condition over that period. In fact, removal of horses and cattle from the adjoining pasture led to the recruitment of shrubs and trees along the forest edges, which increased habitat cover. Poor habitat conditions may be responsible for years of fluctuating estimates at Site 1 due to drought-like conditions. There were 3 years of below average rainfall during 2004–2008 and another 3 years during 2015–2020, which may have destabilized the population given its reliance on eucalypt flowering [43]. This would have been exacerbated by the small and isolated habitat area. 

Whilst we have no direct evidence of increased predation on the squirrel gliders, introduced red foxes (*Vulpes vulples*) have become resident in the area since the first survey period, and domestic cats have also been detected. We are also aware of several breeding pairs of powerful owls (*Ninox strenua*) occupying the habitat within 1–2 km of Site 1, which prey on arboreal mammals such as the greater glider (*Petauroides volans*) and other native possums [44,45,46]. Predation from powerful owls was not known from the study area in the first period, but it is possible that predation may have increased over the study period. There is no reason to suspect that disease caused the decline as there have been no observations of diseased animals. 

The last potential cause of the decline is inbreeding depression. A genetic study found that Site 1 but not Site 2 had the hallmarks of genetic drift [11]. We have not observed any impairment in breeding by female gliders, but it is possible that mortality of juveniles is higher than would otherwise be the case. We were unable to estimate juvenile survival due to the difficulty of targeting this age class. In urban Denmark, a high survival probability for juvenile hedgehogs is critical to maintaining hedgehog populations [47]. Fragmented northern quoll (*Dasyurus hallucatus*) populations in northern Australia are reliant on the survivorship of juveniles [48]. Declines in squirrel glider populations may similarly reflect juvenile mortality. 

Urban populations are frequently small in size due to severe habitat fragmentation [9,49]. These populations are particularly vulnerable to genetic impairment, particularly when geographic isolation limits gene flow [34]. Gardens and green spaces can play vital roles in the preservation of urban fauna by providing habitat and stepping stones that may facilitate gene flow [50,51]. Whether this is relevant for the squirrel glider is yet to be determined. Small populations that show evidence of demographic and genetic degradation may benefit from intervention, such as genetic rescue (i.e., the introduction of genetically diverse individuals to offset the adverse effects of inbreeding and the loss of genetic diversity [52]). 

## 5. Conclusions

Our study has demonstrated different population trajectories in two populations of the squirrel glider, and has provided evidence that the smaller and more isolated population is experiencing long-term decline. The cause of the decline is currently unknown. A longer period of study will be required in order to distinguish year-to-year fluctuations from a longer term sustained decline. 

## Figures and Tables

**Figure 1 animals-13-02098-f001:**
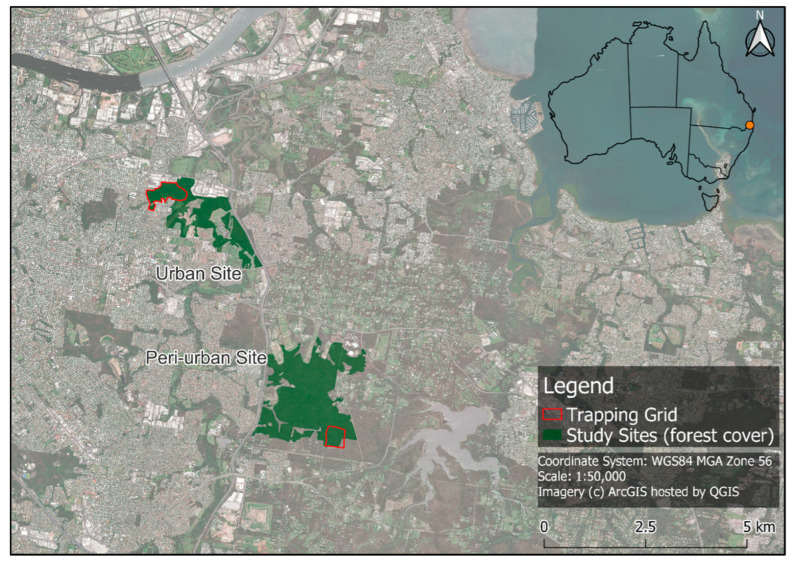
Location of the urban and peri-urban study sites in Southeast Queensland, Australia. The urban matrix surrounding the study sites can be observed through the satellite imagery.

**Figure 2 animals-13-02098-f002:**
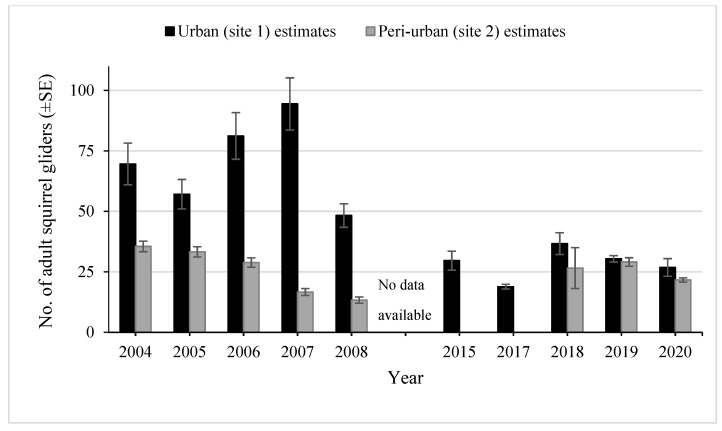
Estimates (±SE) of the number of adult squirrel gliders from 2004 to 2020 within the trapping grids at Site 1 and Site 2.

**Table 1 animals-13-02098-t001:** Comparison of robust design models for adult squirrel gliders captured at Site 1 and Site 2 between 2004 and 2020.

Site	Model ^1^	AICc	ΔAICc	*w*	ML	*K*
Site 1	S (.) p (years reduced) c (years reduced) g’ (0) g” (0)	1831.46	0.00	0.83	1.00	7
	S (.) p (years reduced) c (years reduced) g’ (.) g” (.)	1834.67	3.21	0.17	1.00	9
	S (.) p (years reduced) c (years reduced) g’ (.) g” (.)	1847.43	15.97	0.00	0.00	23
	S (periods) p (years reduced) c (years reduced) g’ (.) g” (.)	1849.53	18.07	0.00	0.00	24
	S (periods) p = c (years reduced) g’ (.) g” (.)	1872.82	41.36	0.00	0.00	14
	S (.) p (.) c (.) g’ (.) g” (.)	1882.18	50.73	0.00	0.00	5
Site 2	S (1) g’ (.) g” (1) p = c (years reduced) yr1 = 2 = 3 = 4 = 5	783.91	0.00	0.92	1.00	6
	S (1) g’ (period) g” (period) p = c yr1 = 2 = 3 = 4 = 5	789.83	5.93	0.05	0.05	9
	S (1) g’ (0) g” (0) p = c (years reduced) yr1 = 2 = 3 = 4 = 5	792.22	8.31	0.01	0.02	5
	S (1) g’ (.) g” (.) p = c (years reduced)	793.08	9.17	0.01	0.01	11

^1^ S = Apparent survival; (.) = value is constant; (years reduced) = the number of sessions (years) were reduced to simplify the model; (period) = parameter was set to estimate the two sample periods, i.e., 2004–2008 and 2015–2020; p = probability of capture; c = probability of recapture; (0) and (1) = parameter was fixed at zero or one; g’ and g” = temporary emigration and immigration.

## Data Availability

Data generated during and/or analyzed for this study can be supplied on reasonable request.

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
