# Peer review of "Are Urban Populations of a Gliding Mammal Vulnerable to Decline?"

_animals, 2023, doi:10.3390/ani13132098_

Round 1

Reviewer 1 Report

Thank you for the opportunity to review this excellent work. I am supportive of its publication after addressing some minor comments:

lines 98-108; please include references; some of these techniques are very important (aging gliders; checking their reproductive status, etc.) and the readers would benefit from citations to these techniques, so as to be able to read more about them if wanting to.  line 163: structured?, is there meant to be '?' after structured? Lines 183-184:  (Fig. 4 错误! 184 未找到引用源。): not sure what should be here; perhaps numbers for references and not the Chinese characters that appear.  Lines 281-284: This paragraph would benefit from elaborating on what genetic rescue is and how it might apply to the Minnippi population.  Lines 293-295: I think this sentence would be better placed in the introduction; my suggestion is to end the conclusion at line 293 '...healthy forests'.  There was no mention of a Qld scientific Licence to collect the data; perhaps need to include this information?  Good luck with your review and I look forward to reading the published version of the manuscript. 

Reviewer 2 Report

Review report “Does urban isolation produce instability in a gliding mammal”. The authors have studied population size of squirrel gliders in two sites, one urban and one in more continuous forest area. They found that population size was larger in urban site, but declined there during the study period. Based on these observations they write much speculation on population dynamics of species in isolated habitats. Starting from title of the work: “urban isolation” this is not really studied. They do not have measured isolation in this study and n=1 for the urban areas; “produce instability” neither this is studied in the current manuscript. Title should be rewritten to more accurately reflect the true content of the work. The same applies to abstract, introduction and discussion.

Do you have some evidence that this urban Minnippi population is somehow isolated or small? How much gliders occur in areas surrounding your trapping site?

Fig 1 is very unclear

At first reading, it was confusing that you use the names Minnippi and Mt Petrie in methods and results. Easier for reader for example, if you call them “urban site” and “forest site” or something like that

It remained unclear whether your model could really separate mortality and immigration. For example, would lower survival in Minnippi be explained by higher immigration away from the site?

Why you have this text in results lines 140-143 on sex ratios? You combined sexes for the analysis and do not present any test for this sex-ratio data.

3.3. Population sizes section could be shortened, because numbers in text partly repeat those in fig 2. No need to explain in text in detail what is presented in figure.

Discussion line 216: “Minnippi showed substantial variation” There is variation (or decline) in Mt Petrie too during the first trapping period.

line 227. For this species, you can also found studies describing no decline

line 243 No need to write about removal of horses from some local pasture

4.3. Genetic implications. Too speculative, you can shortly mention this, but no need to highlight it with a separate section. You do not have any evidence that there is any genetic implications or that those would be reason or result from the observed decline. In fact, many small populations in nature cope very well although genetic variation may be decreased locally

5 conclusions. This whole paragraph should be rewritten to reflect what you actually did here and observed. Everybody knows that long-term studies are needed to study yearly fluctuations. It is not lesson from this study. Multiple populations: you had n=2. Population dynamics in urban setting: you had n= 1. “Loss of species” “healthy forests” not studied (what you did say on this was that your urban sites was better quality line 222!)
